# An allosteric transport mechanism for the AcrAB-TolC multidrug efflux pump

Zhao Wang[1,2†], Guizhen Fan[3†], Corey F Hryc[1,2,4†], James N Blaza[5], Irina I Serysheva[3], Michael F Schmid[1,2], Wah Chiu[1,2,4*], Ben F Luisi[6*], Dijun Du[6*]

[1]National Center for Macromolecular Imaging, Baylor College of Medicine, Houston, United States; [2]Verna and Marrs McLean Department of Biochemistry and Molecular Biology, Baylor College of Medicine, Houston, United States; [3]Department of Biochemistry and Molecular Biology, Structural Biology Imaging Center, The University of Texas Health Science Center at Houston Medical School, Houston, United States; [4]Graduate Program in Structural and Computational Biology and Molecular Biophysics, Baylor College of Medicine, Houston, United States; [5]MRC Mitochondrial Biology Unit, Cambridge Biomedical Campus, Cambridge, United Kingdom; [6]Department of Biochemistry, University of Cambridge, Cambridge, United Kingdom

**Abstract** Bacterial efflux pumps confer multidrug resistance by transporting diverse antibiotics from the cell. In Gram-negative bacteria, some of these pumps form multi-protein assemblies that span the cell envelope. Here, we report the near-atomic resolution cryoEM structures of the *Escherichia coli* AcrAB-TolC multidrug efflux pump in resting and drug transport states, revealing a quaternary structural switch that allosterically couples and synchronizes initial ligand binding with channel opening. Within the transport-activated state, the channel remains open even though the pump cycles through three distinct conformations. Collectively, our data provide a dynamic mechanism for the assembly and operation of the AcrAB-TolC pump.

*For correspondence: wah@bcm.edu (WC); bfl20@cam.ac.uk (BFL); dd339@cam.ac.uk (DD)

†These authors contributed equally to this work

Competing interests: The authors declare that no competing interests exist.

## Introduction

Antibiotic resistance of pathogenic bacteria is a growing clinical problem, exacerbated by insufficient development of new antibiotics. Drug efflux pumps play important roles in intrinsic or acquired drug resistance to a wide variety of currently available antimicrobial agents (*Pu et al., 2016*). In Gram-negative bacteria, some of these pumps form multi-protein assemblies that span the cell envelope (*Du et al., 2015a*). The components of these assemblies include an outer membrane protein, a plasma membrane-spanning protein and a periplasmic protein that connects the two trans-membrane components (*Du et al., 2015b*). AcrAB-TolC is a RND-based tripartite efflux pump, comprised of the outer membrane protein TolC, the periplasmic membrane fusion protein AcrA, and the inner membrane transporter AcrB, which cycles through three different conformational states during the drug transport process: access (L), binding (T) and extrusion (O) (*Du et al., 2015a*). A small peptide, AcrZ, has been identified that modifies the activity of AcrB (*Hobbs et al., 2012*). Previous electron microscopy studies of the AcrAB-TolC pump have revealed the overall shape of the pump and the relative arrangement of its components (*Daury et al., 2016*; *Du et al., 2014*; *Jeong et al., 2016*). Due to the limited resolution, these studies were unable to identify the detailed interaction interfaces between the components. Moreover, the mechanism of pump assembly and channel opening is presently unclear (*Müller and Pos, 2015*; *Zgurskaya et al., 2015*) and requires detailed structural information of the pump in the different conformational states that accompany drug translocation. In

this study, we report the near-atomic resolution cryoEM structures of the AcrAB-TolC pump captured in both resting and drug transport states. The data reveal the detailed tertiary and quaternary structural changes associated with the transition from apo to ligand-bound states. The quality of the cryoEM maps enable unambiguous modeling of all components and identification of the conformational asymmetry in AcrB in the presence of antibiotic that underpins the molecular mechanism of drug translocation. Our study provides a structural framework for understanding the mechanism underlying drug efflux.

## Results

### CryoEM structure determination

Preparation of the intact AcrAB-TolC for structural studies has been a challenge. Previously, the fully-assembled pump was stabilized by tandem fusion and cross-linking of its components using the 'GraFix' method (*Du et al., 2014*). The fusion was co-expressed with AcrZ, a hydrophobic helical peptide that affects AcrB efflux activity for a subset of antibiotics (*Hobbs et al., 2012*) and interacts with AcrB in the transmembrane portion (*Du et al., 2014*). To pursue a higher resolution structure, we improved the purification procedure. The ratio of detergent to total membrane protein was optimized to extract AcrAB-TolC from the cellular membrane, and the detergent was exchanged with amphipol A8-35 (*Baker et al., 2015*; *Liao et al., 2013*; *Popot, 2010*; *Tribet et al., 1996*; *Zoonens and Popot, 2014*). Ligands were used in an effort to favor conformational uniformity and/or to capture structures corresponding to different ligand transport states. The purified pump was imaged in the presence of pyranopyridine inhibitor MBX3132 (*Sjuts et al., 2016*) (*Figure 1—figure supplement 1A and B*; *Supplementary file 1*) or substrate puromycin, as described in Materials and methods.

We also have developed a new procedure to stabilize the full pump using disulphide-linkages that were chosen from the proximity of residues S273 in the β-barrel domain of AcrA and S258 in the β-hairpin motif of the DN subdomain of AcrB seen in the cryo-EM structure of the full pump. We introduced single cysteine-substitutions in the individual AcrA and AcrB components (AcrA-S273C and AcrB-S258C) without flexible engineered linkers. Consistent with the model based on the fusion constructs, we observed disulfide bond formation between the two free components. Moreover, the disulfide bond-stabilized AcrAB can recruit TolC (and three additional AcrAs, which are not expected to be cysteine-crosslinked) to form a stable tripartite complex in vivo (*Figure 1—figure supplement 2A*). This complex was prepared without co-expression of AcrZ. The full pump assembly, stabilized through the disulfide-bond linked AcrAB, was imaged in the absence of ligand (*Figure 1—figure supplement 2, B and C*).

The cryo-EM structures of the apo pump and the pump/ligand complexes reveal different conformational states of the full pump. These structures have been solved at different subnanometer and near atomic resolutions (*Supplementary file 1*). The maps of the full pump at 3.6 Å resolution can be readily segmented into a TolC trimer, a periplasmic AcrA hexamer, and an AcrB trimer (*Video 1*). The corresponding domains and subdomains of the four protein components derived from the cryoEM maps are delineated in *Figure 1—figure supplement 3*. We will describe each structure further below.

**Video 1.** Structure of AcrABZ-TolC multidrug efflux pump with inhibitor MBX3132 bound. Individual protein subunits of the density map are highlighted and their corresponding models are then shown.

### The apo structure of the pump adopts a closed channel

The reconstruction of the apo pump at 6.5 Å resolution (*Supplementary file 1*; *Figure 1—figure supplement 4*) revealed TolC to be in the closed

conformational state (*Figure 1*), like that seen in the crystal structure of the isolated protein (*Koronakis et al., 2000*). The periplasmic component of the pump, AcrA, forms a hexamer through the trimerization of two conformationally distinct protomers (*Figure 1A*). We refer to the two distinct subunits as 'protomer-I' and 'protomer-II'. In both AcrA protomers, a short helix-turn-helix (HTH) motif interacts in a tip-to-tip manner with the HTH motif of the TolC coiled-coils at their periplasmic ends (*Figure 1A*, right panel). Because each subunit of TolC has an internal structural repeat, the trimer has 6 HTH motifs, which interact with the 6 HTH motifs of the AcrA hexamer in a quasi-equivalent manner: AcrA protomer-I and the adjacent protomer-II interdigitate with the intra- and inter-protomer grooves of TolC, respectively. The two distinct conformations of the HTH motif loops of AcrA protomers I and II structurally conform to the slightly different contact surfaces of TolC (*Figure 1A*, right panel). This interaction is consistent with functional (*Lee et al., 2012*; *Song et al., 2014*; *Xu et al., 2010*) and structural data (*Daury et al., 2016*; *Kim et al., 2015*), but differs in detail

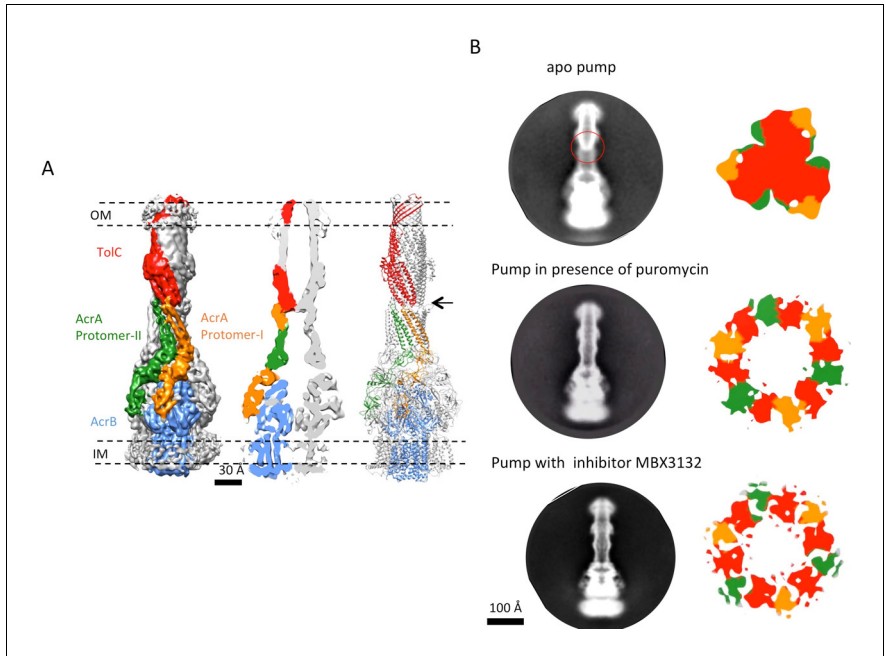

**Figure 1.** The structure of disulfide-bond stabilized AcrAB-TolC pump with closed channel at 6.5 Å resolution. (**A**) CryoEM map of the pump with closed-state TolC, visualized in side view along the membrane plane (left). The four components within one asymmetric unit of the C3 symmetric assembly are color-coded: TolC (red), AcrA (orange and green) and AcrB (blue). (middle) A sliced view of the pump shows the closed channel. (right) Ribbon representation of the pump with closed-state TolC with the same color code as in left panel. The black arrow indicates the closed site in TolC. (**B**) Comparison of reference- free 2D averages (left) and slice view of 3-D maps at the arrow (right) of the apo pump with closed TolC (top), pump in presence of puromycin (middle) and pump with MBX3132 (bottom).

The following figure supplements are available for figure 1:

**Figure supplement 1.** CryoEM structure of the AcrABZ-TolC pump with MBX3132 bound.

**Figure supplement 2.** Analysis of the disulfide-bond stabilized AcrAB-TolC pump.

**Figure supplement 3.** The domains and subdomains of the components of AcrABZ-TolC pump.

**Figure supplement 4.** Resolution estimation of the apo AcrAB-TolC pump with closed TolC.

**Figure supplement 5.** Conformational difference between two AcrA protomers and their interaction with closed-state TolC in the apo-state.

from that proposed for the open state of the pump reported earlier (*Jeong et al., 2016*), as we will describe further below. A gap is present at the interfaces between adjacent AcrA dimer pairs, so that the AcrA helical hairpin, lipoyl and β-barrel domains do not pack tightly to seal the channel from the periplasm (*Figure 1—figure supplement 5*). This gap must be closed in the transport process to prevent leakage of substrates into the periplasm. AcrB adopts a symmetric state (LLL) in the pump assembly, as seen in the crystal structure of apo AcrB (*Murakami et al., 2002*).

## The transport state of the full pump with opened channel

The crystallographic studies of AcrB in complex with natural substrates reveal asymmetric structures, which suggest consecutive steps of a transport cycle (*Murakami et al., 2006*; *Seeger et al., 2006*). We explored how the full pump accommodates AcrB in this functional state in the presence of puromycin, which is a validated transport substrate (*Hobbs et al., 2012*).

We targeted the AcrB protomers using a focused classification procedure (*Bai et al., 2015*) (See Materials and methods) and could identify a subset of the full pump assembly (30.2% of the entire dataset) containing exactly one protomer whose density matches the extrusion state (O) of AcrB, as shown in *Figure 3—figure supplement 1*. A 5.9 Å density map of the full pump assembly without any imposed symmetry was reconstructed from this particle subset (*Figure 2A*; *Figure 2—figure supplement 2*). The correlation analysis of the map region corresponding to the AcrBZ components

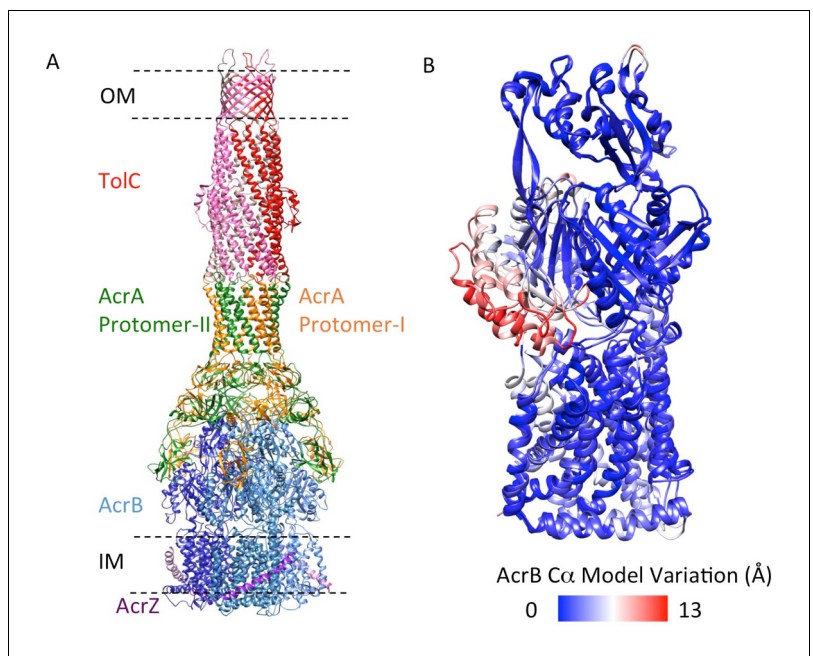

**Figure 2.** 5.9 Å resolution cryoEM asymmetric structure of the AcrABZ-TolC pump in the presence of puromycin. (**A**) Structure of asymmetric AcrABZ-TolC pump visualized in side view; subunits are color-coded accordingly: TolC (red and pink), AcrA (orange and green), AcrB (blue) and AcrZ (purple). (**B**) Model variation of the C-alpha trace between the three subunits of AcrB within the full pump colored in gradient from blue to red with increasing variation.

The following figure supplements are available for figure 2:

**Figure supplement 1.** Workflow of masked classification of C3-symmetry imposed map of AcrABZ-TolC pump in the presence of puromycin with residual signal subtraction.

**Figure supplement 2.** CryoEM analysis of asymmetric AcrABZ-TolC pump in presence of puromycin reconstructed from a subset of particle images after focus classification.

**Figure supplement 3.** Crystal structure of AcrBZ complex with puromycin bound (PDB ID 5NC5).

shows that the three AcrB protomers in L, O and T states are in a fixed spatial relationship with respect to each other in this full pump assembly (*Figure 2B*). This spatial disposition fits very well to the crystal structures of numerous other drug-bound AcrB structures (*Eicher et al., 2012*; *Murakami et al., 2006*; *Nakashima et al., 2011*). Though the AcrB subunit densities match well with those in the crystal structure, no puromycin density is detected in the cryoEM map. A similar situation occurs for some crystal structures of the asymmetric state, where drug is present in the buffer but discernable electron density for the drug is absent from the ligand binding pockets (*Seeger et al., 2006*). The crystal structure of AcrBZ in the presence of puromycin at 3.2 Å (*Figure 2—figure supplement 3*; *Supplementary file 3*) shows that the AcrB subunits adopt three distinct conformational states (*Figure 2—figure supplement 3B*) as seen in our cryoEM structure in the presence of this drug. Poorly defined density for puromycin is found in the binding pocket of the T protomer (*Figure 2—figure supplement 3C*). In the case of the cryoEM map, the lack of density can be rationalized as either a resolution limit in the C1 map or the drug being bound only transiently and flexibly.

TolC adopts a fully opened state via tip-to-tip interactions with AcrA in the pump assembly and AcrAs pack tightly to form a sealed channel, which is similar to that seen in the complex of the pump with the potent inhibitor MBX3132, to which we now turn.

## The structure of pump in complex with inhibitor MBX3132 reveals the interaction interfaces between pump components at atomic resolution

The inhibitor MBX3132 likely binds AcrB tightly, since it is active in the nanomolar concentration range (*Sjuts et al., 2016*), and the compound was used to lock the pump in a more homogeneous conformation. The 3D map for the pump/MBX3132 complex was initially generated without any imposed symmetry and exhibited an apparent 3-fold symmetric pattern (*Figure 1—figure supplement 1C*). Thus, refinements were performed with 3-fold symmetry, which produced the final 3D reconstruction at ~3.6 Å resolution. This dataset was further subjected to focused 3D classification (*Bai et al., 2015*) of a targeted region of AcrB to reveal its structural variations in response to the presence of the inhibitor. We found that 73% of the particles within the dataset have their three

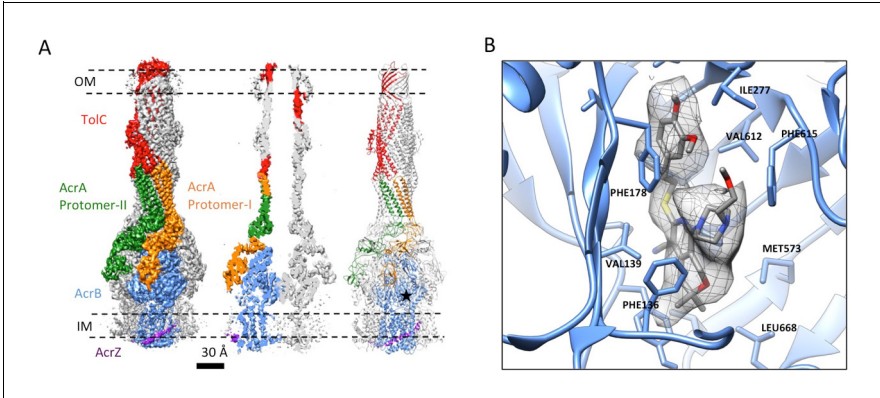

**Figure 3.** 3.6 Å resolution cryoEM structure of the AcrABZ-TolC pump with inhibitor MBX3132 bound AcrB in the TTT state. (**A**) Three-fold symmetry imposed density map (left) and model (right) of the AcrABZ-TolC complex. (middle) A sliced view of pump. The five protein components within one asymmetry unit of the pump are color-coded. TolC (red), AcrA (orange and green), AcrB (blue) and AcrZ (purple). (**B**) Density of the MBX3132 seen in the hydrophobic trap of each AcrB subunit. The location of the trap is indicated by the star in the model in the right panel of (**A**).

The following figure supplements are available for figure 3:

**Figure supplement 1.** Workflow of focused classification with residual signal subtraction for the AcrABZ-TolC/MBX3132 pump reconstructed with C3 symmetry imposition.

**Figure supplement 2.** Validation of AcrABZ-TolC/MBX3132 density map and model.

AcrB subunits in the same (TTT) conformation (Materials and method and *Figure 3—figure supplement 1A–E*). Subsets of particles show AcrB conformations in LLL (1.4%), LLT (6.2%) and LTT (18.6%) states, respectively, while none of the subunits was identified in an 'O' state. Therefore, this inhibitor-bound pump conformation primarily represents a T-saturated state of the AcrB. The C3 symmetry imposed reconstruction using the subset of particles with TTT state (24,597 particles) also achieved 3.6 Å resolution (*Figure 3A*, *Figure 3—figure supplement 1F and G*). We observed clear density for the inhibitor in a 'hydrophobic trap' site adjacent to the drug binding of the AcrB trimer in both the asymmetric and symmetry imposed maps (*Figure 3B*). Side chains can be readily assigned and modeled in the protruding densities along the peptide backbone (*Figure 3—figure supplement 2*). This cryo-EM structure of pump/MBX3132 enables us to visualize the interaction interfaces between the pump components at atomic resolution.

Comparing the structure of TolC in the crystal or in the apo pump with that in the inhibitor-bound pump assembly reveals that TolC adopts a fully opened state through an iris-like dilation of the periplasmic end. This dilation is presumably necessary to maintain TolC-AcrA connectivity and to permit drug molecules to pass through the pump (*Koronakis et al., 2000*). Little conformational change is shown in the TolC TM region and the adjacent α-helical portion. However, the structures start to deviate at the equatorial domain, where the superhelical trajectories of coiled-coil helices change

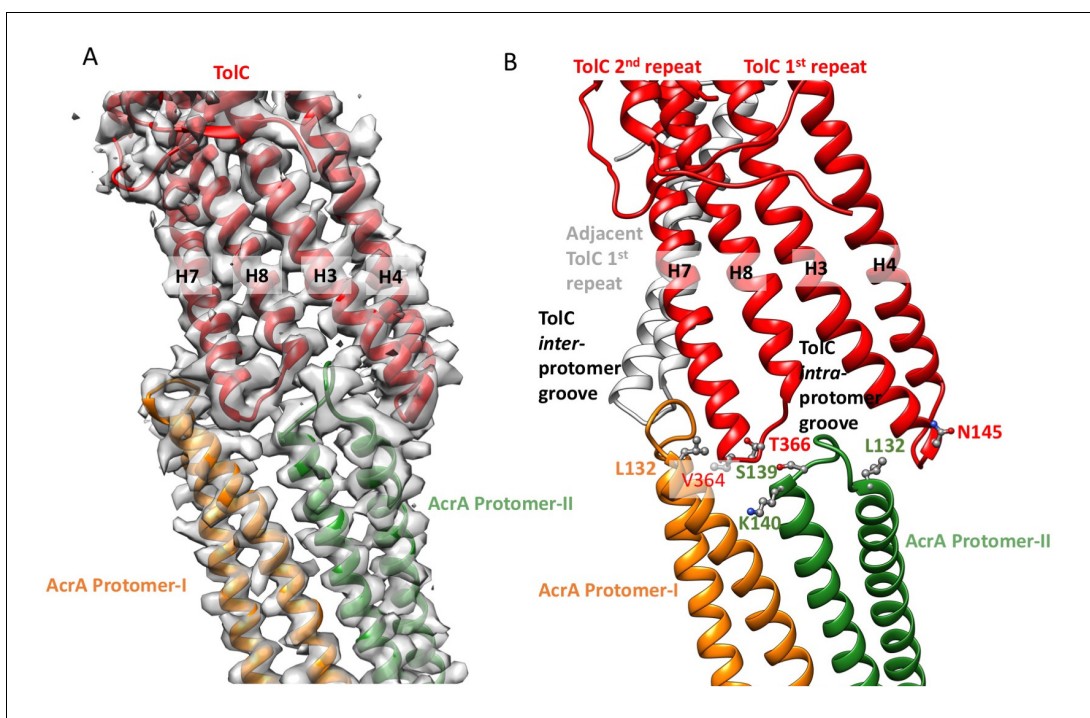

**Figure 4.** Interactions between TolC and AcrA in the AcrABZ-TolC pump with inhibitor MBX3132 bound. (**A**) Segments of the 3.6 Å resolution cryoEM density map of the pump with fitted model showing the tip-to-tip interaction between a TolC protomer (red) and two α-helical hairpins of AcrA (orange and green). (**B**) Detail of the tip-to-tip interface of TolC and AcrA. Residue pairs of complementary mutations (*Kim et al., 2015*) identified at this interface are shown (*Xu et al., 2011*).

The following figure supplements are available for figure 4:

**Figure supplement 1.** Residue interactions in the closed and open states of TolC.

**Figure supplement 2.** Pairs of co-evolved residues at the TolC-AcrA interface.

**Figure supplement 3.** The conformational flexibility at the tips of α-helical hairpin domains of AcrA, displayed as Cα traces.

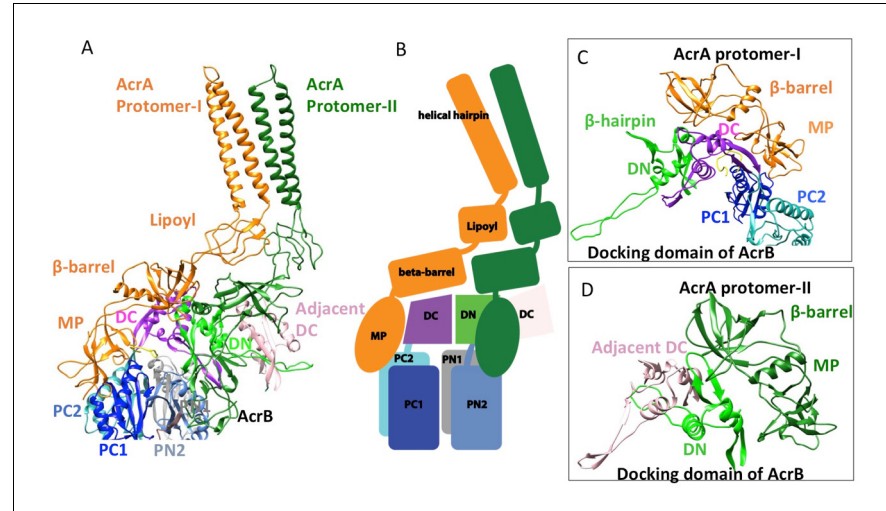

**Figure 5.** Interactions between AcrA and AcrB in the AcrABZ-TolC pump with inhibitor MBX3132 bound. (**A**) A pair of AcrA protomers interacts with one AcrB protomer. Subdomains of the AcrB are labeled with different colors. (**B**) Cartoon schematic of the domains/subdomains shown in (**A**). (**C**) Interaction between AcrA protomer-I and AcrB. (**D**) Interaction between AcrA protomer-II and AcrB. There are 120 and 180 degrees rotations in (**C**) and (**D**) relative to (**A**), respectively.

with the iris-like dilation (*Figure 4A and B*; *Figure 4—figure supplement 1A and B*). In the 'closed state' conformation, seen in the apo pump, the constriction is maintained by an inter-protomer hydrogen-bonding network, involving R367 in one subunit and T152, D153 and Y362 in the adjacent subunit (*Andersen et al., 2002*; *Augustus et al., 2004*; *Bavro et al., 2008*)(*Figure 4—figure supplement 1A*). This network of interactions is broken in the TolC of pump/MBX3132 complex (*Figure 4—figure supplement 1B*) and the pump/puromycin complex. Accordingly, we refer to TolC in the pump-ligand complexes as being in an 'open state' conformation.

We observe tip-to-tip interactions of the AcrA and TolC HTH motifs (*Figure 4*), which is similar to that proposed for the open state of the pump reported earlier (*Jeong et al., 2016*). These interactions account for the importance of the conserved Val-Gly-Leu/Thr element of TolC (*Jeong et al., 2016*; *Song et al., 2014*). At the AcrA protomer-I/TolC interface, contacts involve the backbone of G365 in the conserved VGL motif of TolC with the backbone of K140 and side chain of S139 in AcrA (*Jeong et al., 2016*). The N145 and T366 of TolC directly contact AcrA L132 in the pump, consistent with the results of in vivo site-specific cross-linking experiments (*Xu et al., 2011*) (*Figure 4B*). Our cryoEM structure shows the presence of residue pairs at the interface between AcrA and TolC (*Figure 4—figure supplement 2*) that were predicted to have co-evolved in the homologous proteins in a different efflux pump MdtNOP (*Harley and Saier, 2000*; *Ovchinnikov et al., 2014*). 3D classification identified a small subset of particles, in which TolC rotates by 60 degrees with respect to AcrA, but the alternative interface still retains the key contacts described above. As observed in the apo state, there are two distinct conformations of the HTH motif loops of AcrA protomers I and II that structurally conform to the slightly different contact surfaces of TolC (*Figure 4*). The flexibility of the HTH is consistent with observations from crystal structures (*Figure 4—figure supplement 3*) (*Mikolosko et al., 2006*).

In the six protomers of AcrA, the lipoyl and β-barrel domains form a stack of two continuous rings. The lipoyl domains have no interaction with either AcrB or TolC, and the β-barrel domains contact only the docking domain of AcrB (*Figure 5*). The membrane-proximal (MP) domain of AcrA interacts with both the docking domain and the top of the pore domain of AcrB. Each AcrB protomer makes two non-equivalent interactions with AcrA (*Ntreh et al., 2016*); *i.e.* each pair of AcrA protomers makes two conformationally distinct interfaces with AcrB, as was found with the interaction between AcrA and TolC (*Figures 4* and *5*). AcrA protomer-I makes extensive interactions with AcrB. The β-barrel domain of AcrA protomer-I docks on the DN and DC subdomains of AcrB, while

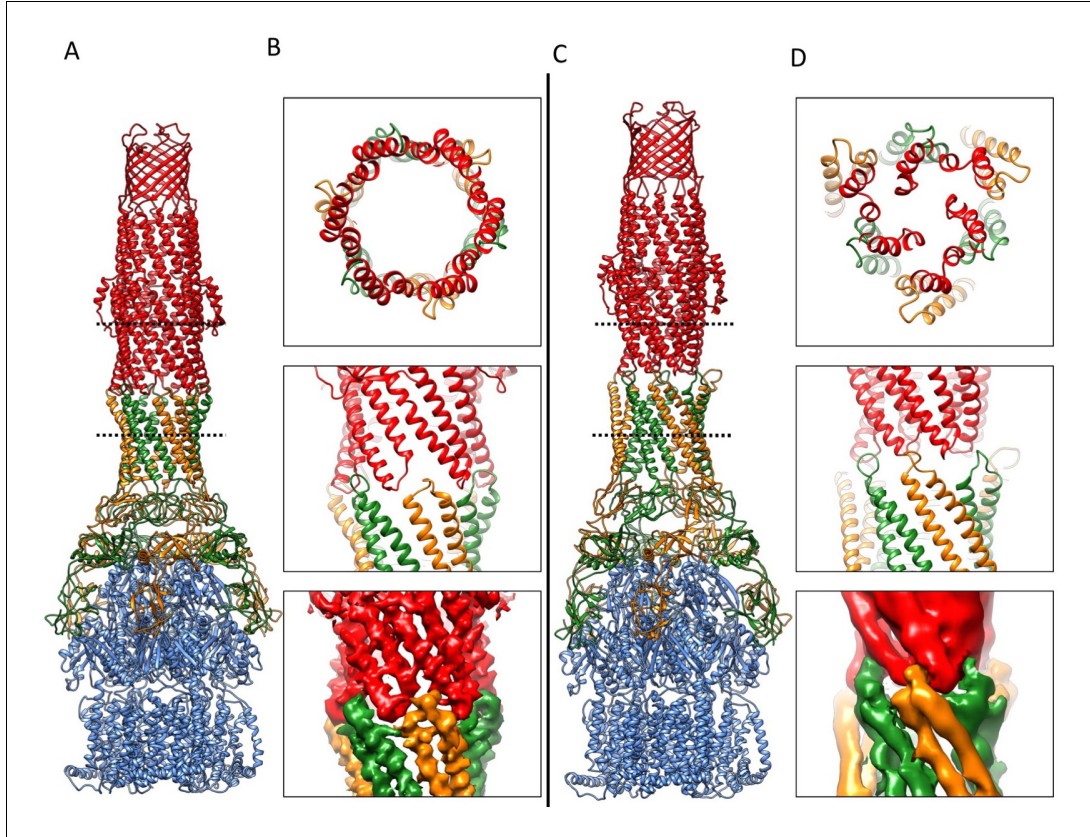

**Figure 6.** Comparison of AcrA-TolC interactions in puromycin-bound and apo state. (**A**) Structure of AcrAB-TolC/puromycin pump visualized in side view. (**B**) Top view (top panel) and side views (middle and bottom panels) of AcrA-TolC interface in puromycin-bound state. (**C**). Structure of apo AcrAB-TolC pump visualized in side view. (**D**) Top view (top panel) and side views (middle and bottom panels) of AcrA-TolC interface in apo state. Subunits are color-coded accordingly: TolC (red), AcrA (orange and green) and AcrB (blue).

its MP domain interacts with the PC1 subdomain, the linker region between PC2 and DC, and an extended loop of the DN subdomain from an adjacent AcrB protomer (*Figure 5A and C*). The spatial orientation of a β-hairpin motif, in the DN subdomain of AcrB, differs between the cryoEM and crystal structures. This conformational difference facilitates the interactions between AcrB and the β-barrel domain of AcrA protomer-II and a short α-helix of the DC subdomain from the adjacent AcrB protomer (*Figure 5A and D*). This contact resembles the interactions observed between the metal transporter subunits CusB and CusA, which are homologs of AcrA and AcrB, respectively (*Su et al., 2011*). This β-hairpin motif in the DN subdomain of AcrB has been shown to be important for the proper assembly of a functioning pump (*Weeks et al., 2014*). The linking region between AcrB subdomains PN2 and DN contacts the end of a β-sheet in the MP domain of AcrA protomer-II (*Figure 5A and D*).

Morph Between
Disulfide-Engineered Pump

Pump in Presence of Puromycin

**Video 2.** Morph animation between the disulfide-engineered pump-derived model and the derived model of the pump in the presence of puromycin. Intermediate states are interpolated between the two resolved states.

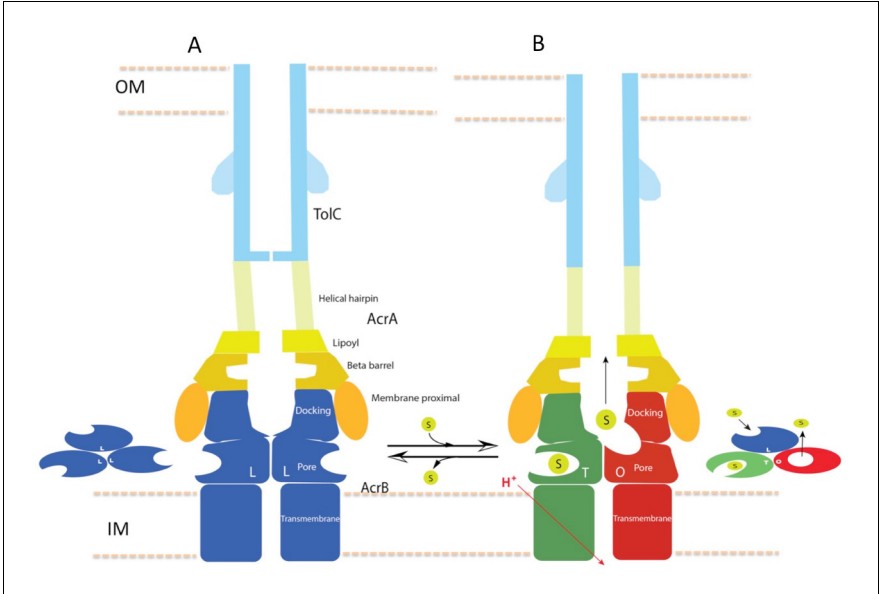

**Figure 7.** Schematic cartoon of the transport mechanism. (**A**) The resting state of the apo pump with TolC in closed-state and the AcrB trimer in LLL conformation. (**B**) The apo pump switches to a transport-state in the presence of transport substrate (s), opening the TolC channel (right arrow). In the transport-state, AcrB cycles through three, structurally distinct states (L, T and O), two of which are shown in the left panel (T and O). Cycling is obligatory for unidirectional transport, driven by coupling with transmembrane proton conduction through the TM domain (red arrow). In the absence of substrate, the pump reverts to the resting state and closes the TolC channel (left arrow). The views are cross-sections through the cell envelope, with only two protomers shown for each of the pump components. The inset cartoons on the left in (**A**) and the right in (**B**) show views down the molecular axis of the AcrB trimer, indicating the states with the configuration inferred from the cryoEM reconstructions. The model predicts a contraction along the long axis of the pump with the switch from apo- to transport-states.

Overlaying the model of the apo pump with the pumps in the presence of ligands reveals profound conformational changes at the AcrA-TolC, AcrA-AcrA and AcrA-AcrB interfaces (*Figure 6*; *Video 2*). The quasi six-fold symmetry of the interface between AcrA helical hairpin and open-state TolC reduces to three-fold symmetry in the apo-form. The rearrangements of AcrA repack the β-barrel and membrane proximal domains in a different orientation on the surface of the AcrB and seal the gaps to the periplasm. Conformational changes in the periplasmic headpiece of AcrB instigate repacking of all four of the AcrA domains, and this causes the reorganization of the coiled-coiled domain of AcrA, which is the key step to organize the HTH motifs and open the TolC channel. Thus, the pump is a highly allosteric system in which conformational changes associated with ligand binding by the apo state of AcrB are communicated over a long distance to TolC through the repacking of protein-protein interfaces (*Fischer and Kandt, 2013*; *Zgurskaya and Nikaido, 1999*).

## Discussion

In this study, we have captured structures of the full pump assembly with a closed channel in the apo-state and with an opened channel in the presence of inhibitor or antibiotic. We propose that the apo-form of the pump with a closed TolC represents a resting state (*Figure 1*), while the pump assemblies with opened TolC in the presence of transportable ligands represent transport states, in which the AcrB trimer adopts asymmetric LLT, LTT or LTO conformations. The three pump components undergo significant conformational changes when the assembly switches from the resting-state to the transport-states. In the transport-states, ligand binding is associated with large conformational changes in AcrB for both PN1/PC2 and PN2/PC1 structural modules that form the drug-binding pocket (*Eicher et al., 2014*) (*Figure 3B*). However, the interfaces between TolC and AcrA, and between AcrA and AcrB do not change significantly in the different transport states (LTO, LTT, LLL).

We propose that ligand binding in the apo state instigates quaternary structural changes in AcrB that are communicated to AcrA, which in turn repacks to trigger tertiary structural changes in TolC that open the channel from a sealed resting state. The repacking to AcrA is critical to seal the gaps in the pump, which would otherwise allow substrate to leak into the periplasm. Thus, the long-distance allosteric coupling between AcrB and TolC, mediated through AcrA, ensures that TolC channel opening is synchronized with closing the assembly to the periplasm. When substrate is absent, the pump assembly returns to its resting state. The above conformational changes are accompanied by a contraction of the pump along the long axis in its transport state by nearly 10 Å (*Figure 7*). We anticipate that the transition from the resting state to the transport states will entail a local compression of the periplasm to accommodate the axial contraction of the pump, and local curvature of the outer membrane and inner membrane near the portal of AcrB. These changes may have impact on the energetics and kinetics of the transport process.

RND-type tripartite multidrug efflux pumps in Gram-negative bacteria are one of the major contributors to multidrug resistance. Efflux pump inhibitors could be used as adjunct therapies to increase the potency of existing antibiotics and counter the emergence of multidrug resistant bacteria. Efforts have been focused on the development of inhibitors for the transporter components. Due to the capacity of the transporters to recognize a broad range of substrates, such compounds are particularly difficult to identify (*Opperman and Nguyen, 2015*). In the presence of inhibitor MBX3132, the AcrB trimer is predominantly trapped in the symmetric TTT conformation (*Figure 2—figure supplement 1E*), possibly because the inhibitor has high affinity to AcrB and cannot be transported. Thus, the cryoEM structure suggests that inhibitors like MBX3132 could effectively inhibit AcrAB-TolC efflux through saturated binding and blocking all sites and prevent AcrB from cycling through different states. It may be possible to block drug translocation by targeting the assembly or conformational switching of the tripartite pumps themselves. In this regard, the interaction interfaces between components, which are well resolved in our cryoEM structure, may be important drug-targeting sites to inhibit the assembly of this and other tripartite pumps.

## Materials and methods

### Construction of vectors for overexpression of disulfide-bond stabilized AcrAB–TolC complex

The cysteine-substitutions were introduced for residues AcrA_S273 and AcrB_S258 into the individual components by site-directed mutagenesis using plasmid pAcBH as a template and primers AcrA$_{S273C}$_F: 5'- GAT CAG ACC ACT GGG TGT ATC ACC CTA CGC GCT ATC ttc-3'/AcrA$_{S273C}$_R: 5'-GAA GAT AGC GCG TAG GGT GAT ACA CCC AGT GGT CTG ATC-3', AcrB$_{S258C}$_F: 5'-GTG AAT CAG GAT GGT TGT CGC GTG CTG CTG CGT GAC-3' /AcrB$_{S258C}$_R: 5'-GTC ACG CAG CAG CAC GCG ACA ACC ATC CTG ATT CAC-3', resulting in the construct pAcBH-*AcrA$_{S273C}$AcrB$_{S258C}$*. The tolC gene was amplified using primers TolCinf_F: 5'-AAG GAG ATA TAC ATA TGA AGA AAT TGC TCC CCA TTC TTA TCG GCC-3' and TolC1392inf_R: 5'-TTG AGA TCT GCC ATA TGT CAA TCA GCA ATA GCA TTC TGT TCC GGC GT-3'. The PCR product was then inserted into the NdeI site of pRSFDuet-1 using the In-Fusion cloning method (Clontech), generating the construct pRSFDuet-1-*tolC1392*.

### Protein preparation

The components of AcrABZ-TolC pump were co-expressed and the cellular membrane was prepared as described previously (*Du et al., 2014*). The purification procedure was optimized to improve the homogeneity and stability of the protein complex. 3.5 g of cellular membrane was re-suspended in 50 ml lysis buffer (20 mM Tris pH 8.0, 400 mM NaCl) and was solubilized with 1.5% DDM. The AcrABZ–TolC complex purified by nickel affinity chromatography was loaded onto a Superose 6 column equilibrated with GF buffer (20 mM Tris pH 8.0, 400 mM NaCl, 0.03% DDM). Fractions containing purified AcrABZ–TolC complex were pooled and concentrated to 0.1 mg ml$^{-1}$ using a Vivaspin column (MWCO: 100 kDa). Amphipol A8-35 (100 mg ml$^{-1}$) was mixed with the protein solution with a mass ratio of amphipol A8-35 to protein of 4:1. The mixture was incubated at 4°C for 3 hr. Polystyrene beads (Bio-Beads SM-2) were then added to the protein/DDM/amphipol A8-35 mixture with a mass ratio of Bio-Beads SM2 to detergent of 10:1. The mixture was gently rotated at 4°C overnight

to remove DDM. The detergent-exchanged AcrABZ–TolC complex was concentrated to 2 mg ml$^{-1}$ using a Vivaspin column (MWCO: 100 kDa). Puromycin (100 mM) or pyranopyridine inhibitor MBX3132 (50 mM) was added to the protein sample at a final concentration of 1 mM or 0.4 mM, respectively, and the mixture was incubated on ice for 1 hr before embedding in vitreous ice.

The disulfide-bond stabilized AcrAB–TolC complex was overexpressed and purified as follows: Genes encoding AcrA and AcrB were deleted from the chromosome of *Escherichia coli* strain C43 (DE3). The resulting C43 (DE3) Δ*acrAB* strain was transformed with plasmids pAcBH-*AcrA$_{S273-C}$AcrB$_{S258C}$* and pRSFDuet-1-*tolC1392*. The culture was grown in 2xYT medium with 100 mg ml$^{-1}$ carbenicillin and 50 mg ml$^{-1}$ kanamycin at 37°C until the culture reached an absorbance, at 600 nm, of 0.5 and was then induced by the addition of 0.1 mM isopropyl 1-thio-*β*-D-galactopyranoside (IPTG) at 25°C overnight. Cells were harvested by centrifugation, and pellets from 10 L culture were re-suspended in 150 ml of lysis buffer (400 mM sodium chloride, 20 mM Tris-HCl, pH: 8.0) with 1 tablet per 50 ml EDTA free protease inhibitor cocktail tablets, 5 U ml$^{-1}$ DNase I and 5 mg ml$^{-1}$ lysozyme, and the mixture was stirred at 4°C for 1 hr to digest the cell wall. The cells were lysed by eight passages through a high-pressure homogenizer (EmulsiFlex) at 15,000 psi. Cell debris was pelleted by centrifugation at 9000 g for 30 min. Cellular membrane was pelleted by ultracentrifugation at 125,755 g for 3 hr. 3.5 gram of membrane pellets were re-suspended in 50 ml of lysis buffer with protease inhibitors and were solubilized by adding 1.5% DDM and stirring at 4°C for 3 hr. Debris was pelleted by ultracentrifugation at 125,755 g for 30 min. Imidazole was added to the membrane solution to a final concentration of 15 mM. Histidine-tagged AcrAB-TolC complex was purified by nickel affinity chromatography using a HiTrap 1 ml chelating column (GE Healthcare Life Sciences) equilibrated with GF buffer (400 mM sodium chloride, 20 mM Tris-HCl, pH 8.0, 0.03% DDM) containing 20 mM imidazole. The column was washed with 50 mM imidazole added to GF buffer. Purified AcrAB-TolC complex was eluted with 500 mM imidazole in GF buffer, concentrated and loaded onto a Superose 6 column equilibrated with GF buffer. Fractions 11–15 containing purified AcrAB-TolC complex were pooled and concentrated to 1 ml using a Vivaspin concentrator (MWCO 100 kDa). Amphipol A8-35 (100 mg/ml) was added to a final concentration of 10 mg/ml and the final volume was adjusted to 2 ml using lysis buffer. The mixture was incubated on ice for 3 hr. 250 mg of Polystyrene beads (Bio-Beads SM2) was then added to the protein/DDM/Amphipols A8-35 mixture and rotated at 4°C overnight to remove DDM. The mixture was loaded onto a mini chromatography column to remove the Polystyrene beads (Bio-Beads SM2). The detergent-exchanged AcrAB–TolC complex was concentrated to 2 mg ml$^{-1}$ using a Vivaspin column (MWCO: 100 kDa) before embedding in vitreous ice.

## Crystallization of AcrBZ–DARPin

The AcrBZ–DARPin complex has been found to yield well diffracting crystals in the presence of antibiotics. The complex was purified as described previously (*Du et al., 2014*). The protein was diluted to 10 mg ml$^{-1}$ using sample buffer (10 mM HEPES pH: 7.5, 50 mM sodium chloride, 0.03% DDM) and was incubated with 1 mM of puromycin for 3 hr at 4°C before crystallization trials. The crystals of AcrBZ–DARPin complex were grown at 20°C using the hanging-droplet vapor diffusion method by mixing 4 µl of protein with 2 µl of reservoir solution (80 mM Bis-Tris, pH 6.0, 50 mM sodium citrate, 120 mM KCl, 10% PEG 4000, 0.5% *N,N*-dimethyldodecylamine *N*-oxide). 200 µl of oil (mixture of 40% silicon oil and 60% paraffin oil) was applied over 1 ml reservoir solution to control the rate of vapor diffusion. Crystals appeared 24 hr after setting up the crystallization trials and reached maximal size in 1 week. The crystals were transferred briefly into reservoir solution supplemented with 25% v/v glycerol as cryo-protectant before flash freezing in liquid nitrogen.

## Crystallographic data collection and structure refinement

The AcrBZ–DARPin complex in the presence of puromycin crystallized in space group P2$_1$2$_1$2$_1$. Data sets were collected using beamline I24 at the Diamond Light Source. The diffraction data were processed using iMosflm (*Battye et al., 2011*) and scaled using SCALA (*Evans, 2006*). As the crystals are sensitive to radiation damage, diffraction data of contiguous 15-degree wedges were collected from multiple crystals and merged to obtain a full dataset. Structures were solved by molecular replacement using Phaser with AcrB–DARPin complex (PDB accession: 4DX5) as a search model and refined using PHENIX and REFMAC5 (*Murshudov et al., 1999*). Coot was used for modeling (*Emsley et al.,*

*2010*). Maps calculated from molecular replacement using the AcrB-DARPin complex revealed clear electron density for AcrZ. Data collection and refinement parameters are presented in *Supplementary file 3*. Structures are shown in *Figure 2—figure supplement 3*.

## Electron cryo-microscopy

For the AcrABZ-TolC/Puromycin sample, a 2.0 µl aliquot at 2 mg/ml was applied onto holey carbon film supported by a 200-mesh R2/1 Quantifoil grid (Quantifoil) that had been previously washed and glow discharged. The grid was blotted and rapidly frozen in liquid ethane using a Vitrobot IV (FEI) with constant temperature and humidity. The grid was stored in liquid nitrogen before imaging. Images of frozen-hydrated AcrABZ-TolC/puromycin particles were acquired on a FEI Tecnai G2 Polara electron microscope (FEI) operated at 300 kV using a K2 Summit direct electron detector camera (Gatan).

For the samples AcrABZ-TolC/MBX3132 and apo AcrAB-TolC, a 3 µl aliquot at a concentration of 2 mg ml$^{-1}$ was applied onto glow-discharged holey carbon grid (Quantifoil Au R1.2/1.3, 300 mesh). The grid was blotted and flash frozen in liquid ethane using a Vitrobot IV (FEI) with constant temperature and humidity. The grid was stored in liquid nitrogen before imaging. Zero-energy-loss images of frozen-hydrated AcrABZ-TolC/MBX3132 or apo AcrAB-TolC particles were recorded automatically on an FEI Titan Krios electron microscope at 300 kV, using a slit width of 20 eV on a GIF Quantum energy filter and a Gatan K2-Summit direct electron detector.

The data collection parameters for all three specimens are summarized in *Supplementary file 1*.

## Image processing and 3D reconstruction

For the AcrABZ-TolC/puromycin pump, dose-fractionated super-resolution raw image stacks were binned 2 × 2 by Fourier cropping resulting in a pixel size of 1.62 Å for further image processing. Each image stack was subjected to motion correction using *dosefgpu_driftcorr* (*Li et al., 2013*), and a sum of sub-frames 1–29 in each image stack was used for further image processing. The signal in the motion-corrected images extends beyond 4 Å. Defocus and astigmatism were determined for each micrograph by CTFFIND3 (*Mindell and Grigorieff, 2003*). Each image was binned 2 × 2 to enhance image contrast for particle picking. 99,385 particles were boxed out manually from 6456 micrographs using *e2boxer.py*. An initial map was generated with 3-fold symmetry imposition from 2-D reference-free averages using EMAN2 (*Tang et al., 2007*). The initial reference map was low pass filtered to 60 Å resolution and was used as a starting point for the RELION-1.4 refinement (*Scheres, 2012*). The first round of refinement resulted in a sub-nanometer resolution map. After this step, several rounds of iterative 3D classification and 3D auto-refinement were run to extract a self-consistent subset of the particle data. 67,436 particles were selected after 3D classification and further refined with averages from sub-frames 2–16 to achieve 3.9 Å resolution based on the gold standard criterion (*Henderson et al., 2012*). The final refinement was done using oversampling by a factor of two on the whole dataset. A soft mask in RELION post-processing was applied before computing the FSCs. The final resolution was estimated by 0.143 cutoff of FSCt. Local resolution variations were estimated with ResMap using the two independent maps (*Kucukelbir et al., 2014*).

For AcrABZ-TolC/MBX3132, the software MotionCorr (*Li et al., 2013*) was used for whole-frame motion correction, Ctffind4 (*Rohou and Grigorieff, 2015*) for estimation of the contrast transfer function parameters, and RELION-1.4 package for all other image processing steps. For apo AcrAB-TolC, the software MotionCor2 (*Zheng et al., 2016*) was used for whole-frame motion correction and dose weighting, Gctf (*Zhang, 2016*) for estimation of the contrast transfer function parameters, and RELION-2.0/beta package for all other image processing steps. A particle subset was manually selected to calculate reference-free 2D class averages, which was then used as templates for automated particle picking of the entire data set. The templates were lowpass filtered to 20 Å to limit model bias. Then initial runs of 2D and 3D classifications were used to remove the heterogeneous particles, as well as the false positive particles from the auto-picking. We selected good particles for further analysis based on the quality and high resolution in the 2D and 3D classification.

For the AcrABZ-TolC/MBX3132 pump, 65,256 particles were picked automatically from a total of 1150 micrographs. After initial 2D and 3D classifications, a homogenous subset of 33,587 particles was selected for a first 3D auto-refinement, generating a reconstruction with a resolution of 6.67 Å. After per-particle motion correction and radiation-damage weighting, the polished particles were

submitted to a second round of 3D auto-refinement by applying a soft mask around the TolC, AcrA and the periplasmic headpiece of AcrB (*Scheres, 2016*). These polished particles gave a reconstruction with a resolution of 3.6 Å based on the Gold-standard FSC 0.143. In the case of apo AcrAB-TolC pump, 95,410 particles were picked automatically from a total of 2292 micrographs. After initial 2D classification and two rounds of 3D classifications, 13,544 homogeneous particles were selected for 3D auto-refinement, which generated a map with a resolution of 6.5 Å based on the Gold-standard FSC 0.143. 3D classification also identified a small subset of particles, in which TolC is rotated by 60 degrees with respect to AcrA.

All 3D classifications and refinements were started from a 50–60 Å low-pass filtered initial model, the first of which was made from our previous 16 Å resolution map. The density map was sharpened by applying a negative B-factor estimated by automated procedures (*Rosenthal and Henderson, 2003*). Local resolution variations were estimated using ResMap and visualized with Chimera (*Pettersen et al., 2004*).

## Symmetry release and focused classification

The density map generated from RELION was used to assess the structural asymmetry of the pump by performing focused classification. A soft mask of the targeted region of this map was generated using *relion_mask_create* with a soft edge extension of 5 pixels (*Figure 2—figure supplement 1A*; *Figure 3—figure supplement 1A*). We used this masked volume to set to zero the density of the targeted region from the 3D map of the whole pump, and generated a modified map (*Figure 2—figure supplement 1B*; *Figure 3—figure supplement 1B*). This altered density map was rotated by 120 and 240 degrees about the molecular three-fold axis. The three resulting 3D density maps represent the entire pump with the deletion of each of the three-targeted regions (i.e. each of the three subunits of AcrB) (*Figure 2—figure supplement 1C*; *Figure 3—figure supplement 1C*). Next, each of these three altered density maps was projected in 2D in the corresponding orientations of each of the raw particle images and subtracted from the original image (*Figure 2—figure supplement 1C*; *Figure 3—figure supplement 1C*). Altogether, the particle images (33,337 for the pump with MBX3132 and 67,436 for the pump in presence of puromycin) used in the final reconstruction generated three times that many subtracted particle images. Each of the three subtracted projections from an individual particle image (*Figure 2—figure supplement 1C*; *Figure 3—figure supplement 1C*) was assigned appropriate Euler angles so that they are all in the same orientation in the 3D map to permit 3D classification (*Figure 2—figure supplement 1D*; *Figure 3—figure supplement 1D*). The soft mask generated in the beginning of this process (*Figure 2—figure supplement 1A*; *Figure 3—figure supplement 1A*) was used for the 3D classifications to remove density outside the mask. We ran rounds of 3D classifications using the particle alignment parameters of the symmetry-imposed map of the pump. Each round was iterated 25 times and classified into several subsets (*Figure 2—figure supplement 1D*; *Figure 3—figure supplement 1D*). Focused classification of pump with MBX3132 result in two populations different in the binding site of MBX3132 according to crystal structure (*Figure 3—figure supplement 1D*). Subsets of original and unmodified particle images with different putative ligand binding sites were selected for 3D refinement with no imposed symmetry. We carried out the similar approach on the data of the pump in presence of puromycin. After 3 rounds of 3D classification, we identified two significant conformations. Both class 1 and 2 have a conformation in the pore domain that is closed towards the periplasm. In contrast, the pore domain in class four shows an opened conformation facing the periplasm. We selected the original and unmodified pump particle images that assign only one member from class 1 and 2 for another round of 3D refinement without symmetry constraint (*Figure 2—figure supplement 1E*).

## Model docking and optimization

The crystal structures of trimeric AcrBZ (PDB code: 4C48) and trimeric TolC (PDB code: 1EK9) were docked into the cryoEM map by using Chimera. Both the TolC and AcrBZ models were adjusted manually to optimize the local fit to density using Coot. Individual domains of AcrA from the crystal structure (PDB code: 2F1M) including β-barrel domain, lipoyl domain and α-helical hairpin domain were fitted to the density map using Chimera (*Pettersen et al., 2004*). A homology model of the MP domain of AcrA was built based on the structure of MexA (PDB code: 2V4D) and fit to the density. Four conformationally different structures of the α-helical hairpin domain (Chain A-D in the

crystal structure) (*Figure 4—figure supplement 3*) were evaluated for their fit into the density map of our two protomers. The α-helical hairpin structures from chains A and D fit best into the density map for AcrA protomer-I and for protomer-II in the pump assembly, respectively. The model for the complete pump was optimized using Phenix real-space refinement (*Wang et al., 2014*) with three-fold symmetry imposed (*Supplementary file 2*). The model of the whole pump complex was validated by computing a FSC with the density map. It is 4.1 Å at 0.5 FSC (*Figure 3—figure supplement 2A*). MolProbity (*Chen et al., 2010*) statistics were computed to ensure proper stereochemistry (*Supplementary file 2*).

The 5.9 Å resolution map calculated from the particle subset in class 1 and 2 without symmetry restraint was fit with the symmetric model based on the 3.9 Å density map. We found that TolC and AcrA did not show conformational change. We noticed during the modeling process of TolC alone that the TolC and AcrA interface is polymorphic, with a slight majority of TolC rotated 60 degrees with respect to the dominant configuration seen in the inhibitor bound pump structure, as shown in *Figure 3*.

However, the region for the AcrB in the new map revealed significant mismatch to the symmetric model. A better fit was obtained using the asymmetric AcrBZ crystal structure. Model optimization was then done with Phenix real-space refine (*Wang et al., 2014*) using stronger secondary-structure restraints parameters to maintain proper stereochemistry where weak density exists. Moreover, atomic displacement parameters (B-factors) were computed to determine the level of resolvability throughout the map, which correlated well with Resmap results. The Coot adjustments and Phenix model optimization were iterated one additional round to ensure that the model was a good representation of the map.

For the modeling of the apo state AcrAB-TolC pump, resolved to 6.5 Å resolution, a similar method to modeling of the asymmetric maps was utilized. Crystal structures were rigid-body fit into the density map and model optimization was then carried out with Phenix real-space refine. Again, due to the weaker resolution stronger stereochemical and secondary structure restraints were used to ensure that α-helices and β-sheets did not deviate far from their expected geometry. Manual adjustments were kept to a minimum to reduce human bias in the modeling procedure, with Coot only being used to fix obvious errors such as C-beta deviations. A final check of MolProbity and cross-correlation was done to ensure model quality.

## Accession numbers

Accession codes: cryoEM maps and models of AcrABZ-TolC have been deposited in EMDB under accession code EMD-8636 (apo form), EMD-8640 (puromycin), EMD-3636 (MBX3132 inhibitor), and in the Protein Data Bank under accession code 5V5S(apo form), 5V78 (puromycin), and 5NG5 (MBX3132). The crystal structure of AcrBZ has been deposited to Protein Data Bank under accession code 5NC5.

## Acknowledgements

We thank Matthew L Baker and Grigore D Pintilie for their assistance with modeling and segmentation and to Richard Henderson for helpful discussions. We thank Martin Pos and Timothy Opperman for the reagent MBX3132 and for helpful advice. We thank Satoshi Matsumura for kindly providing the expression plasmid pAcBH. This work was supported by the American Heart Association (16RNT29720001), and Grants from National Institutes of Health (P41GM103832, R01GM079429, R01GM072804, S10OD016279), the Wellcome trust, the Human Frontiers Science Program and MRC Mitochondrial Biology Unit (Grant number: U105663141). CFH is supported by a pre-doctoral training fellowship from the Keck Center of the Gulf Coast Consortia, on the NLM Training Program in Biomedical Informatics (Grant No. T15LM007093). We acknowledge the computing resources provided by the Center for Computational and Integrative Biomedical Research of Baylor College of Medicine and the Texas Advanced Computing Center at the University of Texas at Austin funded by the National Science Foundation (NSF) through grant ACI-1134872.

## Additional information

### Funding

| Funder | Grant reference number | Author |
|---|---|---|
| U.S. National Library of Medicine | Training Program in Biomedical Informatics, T15LM007093 (PI Lydia Kavraki) | Corey F Hryc |
| University of Texas at Austin | ACI-1134872 (PI Daniel Stanzione) | Corey F Hryc |
| Medical Research Council | Mitochondrial Biology Unit, U105663141 (PI Judy Hirst) | James N Blaza |
| American Heart Association | 16GRNT29720001 | Irina I Serysheva |
| National Institutes of Health | R01GM072804 | Irina I Serysheva |
| National Institutes of Health | S10OD016279 | Irina I Serysheva |
| National Institutes of Health | P41GM103832 | Wah Chiu |
| National Institutes of Health | R01GM079429 | Wah Chiu |
| Wellcome | | Ben F Luisi |
| Human Frontier Science Program | | Ben F Luisi |

The funders had no role in study design, data collection and interpretation, or the decision to submit the work for publication.

### Author contributions

ZW, Data curation, Formal analysis, Validation, Investigation, Visualization, Methodology, Writing—original draft, Writing—review and editing; GF, IIS, Data curation, Formal analysis, Funding acquisition, Writing—editing draft; CFH, Data curation, Validation, Visualization, Methodology, Writing—review and editing; JNB, Methodology, Writing—review and editing; MFS, Formal analysis, Validation, Methodology; WC, Supervision, Funding acquisition, Methodology, Writing—original draft, Project administration, Writing—review and editing; BFL, Conceptualization, Data curation, Supervision, Funding acquisition, Writing—original draft, Project administration, Writing—review and editing; DD, Conceptualization, Data curation, Formal analysis, Supervision, Validation, Investigation, Visualization, Methodology, Writing—original draft, Project administration, Writing—review and editing

### Author ORCIDs

Corey F Hryc, http://orcid.org/0000-0002-7277-5249
James N Blaza, http://orcid.org/0000-0001-5420-2116
Ben F Luisi, http://orcid.org/0000-0003-1144-9877

## Additional files

### Supplementary files

• Supplementary file 1. CryoEM Data collection and processing.

• Supplementary file 2 Model statistics of AcrABZ-TolC/MBX3132 pump.

• Supplementary file 3. Crystallographic data and refinement statistics of AcrBZ complex with puromycin.

### Major datasets

The following datasets were generated:

| Author(s) | Year | Dataset title | Dataset URL | Database, license, and accessibility information |
|---|---|---|---|---|
| Wang Z, Hryc CF, Serysheva II, Schmid MF, Chiu W, Luisi BF, Du D, Guizhen Fan, James N Blaza | 2017 | The apo structure of AcrAB-TolC tripartite multidrug efflux pump: 2017 | https://www.ebi.ac.uk/pdbe/emdb/EMD-8636 | Publicly available at EBI Protein Data Bank (accession no: EMD-8636) |
| Wang Z, Hryc CF, Serysheva II, Schmid MF, Chiu W, Luisi BF, Du D, Guizhen Fan, James N Blaza | 2017 | The asymmetric structure of AcrAB-TolC tripartite multidrug efflux pump: 2017 | https://www.ebi.ac.uk/pdbe/emdb/EMD-8640 | Publicly available at EBI Protein Data Bank (accession no: EMD-8640) |
| Wang Z, Hryc CF, Serysheva II, Schmid MF, Chiu W, Luisi BF, Du D, James N Blaza, Guizhen Fan | 2017 | The symmetric structure of AcrAB-TolC tripartite multidrug efflux pump with inhibitor MBX3132 bound: 2017 | https://www.ebi.ac.uk/pdbe/emdb/EMD-3636 | Publicly available at EBI Protein Data Bank (accession no: EMD-3636) |
| Wang Z, Hryc CF, Serysheva II, Schmid MF, Chiu W, Luisi BF, Du D, Guizhen Fan, James N Blaza | 2017 | Multi-drug efflux; membrane transport; RND superfamily; drug resistance | http://www.rcsb.org/pdb/explore/explore.do?structureId=5V5S | Publicly available at PDB (accession no: 5V5S) |
| Wang Z, Hryc CF, Serysheva II, Schmid MF, Chiu W, Luisi BF, Du D, Guizhen Fan, James N Blaza | 2017 | Asymmetric structure of AcrAB-TolC tripartite multi drug efflux pump | http://www.rcsb.org/pdb/explore/explore.do?structureId=5V78 | Publicly available at PDB (accession no: 5V78) |
| Wang Z, Hryc CF, Blaze JN, Serysheva II, Schmid MF, Chiu W, Luisi BF, Du D, James N Blaza, Guizhen Fan | 2017 | Multi-drug efflux; membrane transport; RND superfamily; Drug resistance | http://www.rcsb.org/pdb/explore/explore.do?structureId=5NG5 | Publicly available at PDB (accession no: 5NG5) |
| Du D, Luisi BF | 2017 | Crystal structure of AcrBZ complex: 2017 | http://www.rcsb.org/pdb/explore/explore.do?structureId=5NC5 | Publicly available at PDB (accession no: 5NC5) |

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
