## [Decision Letter]

Thank you for submitting your article "An allosteric transport mechanism for the AcrAB-TolC Multidrug Efflux Pump" for consideration by *eLife*. Your article has been reviewed by three peer reviewers, one of whom, Olga Boudker, is a member of our Board of Reviewing Editors, and the evaluation has been overseen by Richard Aldrich as the Senior Editor. The following individual involved in review of your submission has agreed to reveal his identity: Martin Picard (Reviewer #3).

The reviewers have discussed the reviews with one another and the Reviewing Editor has drafted this decision to help you prepare a revised submission. The reviewers were all very enthusiastic about your work and had suggestions that were editorial in nature:

Reviewer 1:

In the manuscript, the authors present a series of Cryo-EM and crystal structures that illuminate the allosteric coupling in RND family of transporters. They describe the structures of the apo, and the substrate- or inhibitor-bound states of AcrAB-TolC complex. In the apo state, the protein channel that extends through the periplasmic space is closed at the AcrA-TolC juncture. Binding of the substrate or inhibitor to AcrB is coupled to conformational changes through AcrA to the opening of the constriction at AcrA-TolC interface. The presented structures are informative and the work seems solid. I have only comments that are editorial in nature:

1) The physiological role of AcrZ and why it was used in the studies of the substrate-bound pump need to be described in more detail.

2) It is not clear whether there are any structural changes that would propagate through AcrA between substrate- and inhibitor-bound pumps.

3) More detailed figures fro Apo pump showing differences between two conformations of AcrA and their packing with TolC. Also, showing the openings to the periplasm.

4) Show puromycin density in Figure 3—figure supplement 3.

5) Figure 3—figure supplement 3: spelling "deepinding".

6) Show more detailed comparison of AcrA- TolC interactions in ligand-bound and apo state.

7) A bit more detailed description of what are the overall structural changes between closed apo to open substrate-bound state?

8) Switch the order of Figure 2 and 3.

9) Figure 4 could be clearer if the structures in A and B were shown on the same scale and in the same orientation.

10) In Figure 4—figure supplement 1, which structure is shown in B?

11) Subsection “The structure of pump in complex with inhibitor MBX3132 reveals the interaction interfaces between pump components at atomic resolution” paragraph two, did you mean Figure 4—figure supplement 1?

12) Please show in Figure 4 the location of the VGL motif in TolC and the interacting residues of AcrA.

13) In subsection “The structure of pump in complex with inhibitor MBX3132 reveals the interaction interfaces between pump components at atomic resolution” paragraph three what are the residue pairs?

Reviewer 2

The Cambridge group earlier was successful in elucidating, for the first time, the structure of the AcrAB-TolC tripartite efflux complex. Here they report, together with other scientists, on a more refined structure of this complex both in its resting state and in the presumably transport-engaged state, and show how the closed end of the TolC channel may become opened in association of the altered conformation of the pump AcrB. Improvement in technology that led to these wonderful results included many features, such as substitution of Amphipol for conventional detergents and use of focused classification in selecting images. The manuscript is fine. I have only mostly minor comments.

Subsection “CryoEM structure determination” paragraph one. Citing the two papers where amphipol was successfully used would not be enough, especially the journal is targeted toward diverse audience (and also because these two papers do not even cite the original Jean-Luc Popot 1996 PNAS paper on amphipol!).

Paragraph two of the same section. I suppose that disulfide-linked AcrAB recruits not only TolC, but also additional AcrA monomers, as seen in Figure 1—figure supplement 2.

In the last line of paragraph two: I believe that this should be Figure 1—figure supplement 2.

Even with the non-intuitive naming convention of supplementary figures, it does not help that Figure 3 supplement comes before Figure 2 supplement. Please fix this confusing numbering system for the supplementary figures.

Subsection “The structure of pump in complex with inhibitor MBX3132 reveals the interaction interfaces between pump components at atomic resolution”. Inhibitor MBX3132 gave such wonderful results precisely because it binds very tightly to the binding pocket, as seen in the very large negative values of its calculated binding energy in Sjuts et al., 2016 The high affinity of this compound must be mentioned somewhere.

In paragraph three of the same section: conserved Val-Gly-Leu/Thr element in AcrA? It may be obvious, but it would be better to specify AcrA because TolC is also discussed in the same paragraph.

Also in this paragraph, the tip-to-tip interaction of AcrA and TolC HTH motifs is made very clear with Figure 4. However, non-specialists will miss the absence of discussion (and especially figure!) on what happens to these residues in the complex containing a closed TolC conformer. Also, the closed conformer of TolC is supposed to be stabilized by multiple interactions between coiled coils. Are the tip-to-tip interactions enough to overcome these pre-existing multiple interactions?

Paragraph four of the same section and Figure 5. This is very nice and informative, but I miss the comparison with the situation in the apo complex. The next paragraph talks about "overlays", but again there is no figure showing such overlays (especially for the AcrA-TolC interface) even in the supplement.

Discussion section. If TolC-AcrA interface does not change significantly by the binding of the substrate to AcrB, how could the TolC gate become open in the normal operation of the pump? This reminds us that the "open" complex was seen only when MBX3132 was bound to all three of the trimers, and could be an artifact of the experimental setup.

Reviewer 3

The manuscript by Wang et al.et al. describes the near atomic cryoEM structure of the AcrAB-TolC efflux pump from E. coli*E. coli*. Structures of the full pump assembly have been obtained in the apo state and in the presence of a substrate or of an inhibitor. For the first time the authors describe the pump under its closed and open conformations. To do so, they have optimized the biochemical procedure to stabilize the genuine complex, by inserting cysteine residues where the contacts between partners of the pumps are anticipated to take place and by subjecting the pump to disulphide crosslinking. The assembly is stabilized by amphipols. The structures have been obtained by crystallography (AcrBZ in the presence of substrate) and cryoEM (tri-quadripartite structures) using the latest technical developments and methodologies.

This work is a real tour de force and will undoubtedly have a great impact in the field. Hence we fully recommend it to be published in eLIFE*eLife* without further additional experiments to be performed. We only point below minor comments in order for the manuscript to be fully acceptable.

AcrZ seems to be present in the structures obtained in the presence of puromycin and in that in the presence of the inhibitor but absent in the apo states. The authors do not comment upon that point. Could it be that indeed AcrZ is recruited differently whether the substrate is present or not? What could be the possible role of AcrZ on the transport and/or activity?

Note that we advise the authors not to color AcrZ in purple because it is difficult (in particular for a daltonian reviewer) to see it in the very middle of an AcrB molecule colored in blue!

Subsection “The transport state of the full pump with opened channel” first paragraph: Could the authors give the reference to the paper mentioning that puromycin is a validated transport substrate of AcrB?

We believe that here and there, additional papers could be mentioned, see below:

Subsection “CryoEM Structure determination”: There are more and more papers showing that amphipols are indeed very useful in the EM field. The authors refer to the work by Baker, Fan and Serysheva, 2015 and by Liao, Cao, Julius and Cheng, 2013. We suggest to also cite the following review by J-L Popot (Zoonens and Popot, 2014), so that the lab where this molecule was developed is also acknowledged.

Subsection “The structure of pump in complex with inhibitor MBX3132 reveals the interaction interfaces between pump components at atomic resolution”, paragraph two: Maintenance of TolC thanks to the interprotomer network involving R367, T152, D153 and Y362 was already shown thanks to electrophysiology (Andersen et al., 2002), microbiology (Augustus et al., 2004) and crystallography (Bavro et al., 2008).

Paragraph four of the same section: Non equivalent interactions between each AcrB promoter and each pair of AcrA promoter is reminiscent to the work by the laboratory of Helen Zgurskaya on the related pump TriABC-OpmH (Ntreh, Weeks, Nickels and Zgurskaya, 2016)

In paragraph five of the section: Wang et al. comment on the fact that the pump is a highly allosteric system in which conformational changes associated with ligand binding by AcrB are communicated to TolC. Conversely, it was shown that the MFP is mandatory for the activity of the pump (Zgurskaya and Nikaido, 1999), a fact that was nicely discussed by Fischer and Kandt (in section 4.3.3 of Fischer and Kandt, 2013, BBA – Biomembranes 1828, 632-641).

---

## [Author Response]

Reviewer 1:

*[…] 1) The physiological role of AcrZ and why it was used in the studies of the substrate-bound pump need to be described in more detail.*

We have expanded the description in the text about the known function of AcrZ. Regarding why we used AcrZ in the studies of the substrate-bound pump, AcrZ is a modulator of AcrB for the efflux of substrate puromycin, and the his-tagged version of AcrZ helped to purify the assembled pump from the expression host.

*2) It is not clear whether there are any structural changes that would propagate through AcrA between substrate- and inhibitor-bound pumps.*

Within the resolution limit of the substrate- and inhibitor-bound structures, we cannot see significant structural changes propagated through AcrA. In the substrate- and inhibitor-bound pumps, AcrB adopts LTO and TTT states respectively. The interaction interfaces between AcrA and AcrB in these two pump assemblies are slightly different, and the conformation of AcrA protomers is slightly different to adapt to the conformation of AcrB. However, the interfaces between AcrA and TolC are similar. In the case of the switch from apo- to ligand-bound states, the AcrA helical hairpins repack, as a result of changes in the interactions between the AcrA and AcrB. We suggest that these are predominantly quaternary structural changes.

*3) More detailed figures fro Apo pump showing differences between two conformations of AcrA and their packing with TolC. Also, showing the openings to the periplasm.*

We have included a new figure that shows these features (Figure 1—figure supplement 5).

*4) Show puromycin density in Figure 3—figure supplement 3.*

We have now included the puromycin density in the new Figure 2—figure supplement 3 (note that the order of the figures have been changed in the revised text).

5) Figure 3—figure supplement 3: spelling "deepinding".

This is now corrected.

*6) Show more detailed comparison of AcrA- TolC interactions in ligand-bound and apo state.*

The comparison is shown in the Video 2, and in addition, we have included a new Figure 6.

*7) A bit more detailed description of what are the overall structural changes between closed apo to open substrate-bound state?*

We have included a new Figure 6 and reference to Video 2, which help to visualise the overall structural changes between the apo and substrate-bound states.

*8) Switch the order of Figure 2 and 3.*

We have changed the order in the new draft.

*9) Figure 4 could be clearer if the structures in A and B were shown on the same scale and in the same orientation.*

We have revised the figure to place the structures on the same scale.

*10) In Figure 4—figure supplement 1, which structure is shown in B?*

The figure shows the TolC structure in the inhibitor-bound pump (which is very similar to the drug-bound structure). We have modified the figure legend to explain that the structure is for the inhibitor bound state.

*11) Subsection “The structure of pump in complex with inhibitor MBX3132 reveals the interaction interfaces between pump components at atomic resolution” paragraph two, did you mean Figure 4—figure supplement 1?*

Yes, we do. Thank you for spotting this. It has now been corrected in the modified text.

*12) Please show in Figure 4 the location of the VGL motif in TolC and the interacting residues of AcrA.*

These have now been annotated.

*13) In subsection “The structure of pump in complex with inhibitor MBX3132 reveals the interaction interfaces between pump components at atomic resolution” paragraph three what are the residue pairs?*

The pairs are co-evolving residues, and these are listed in Figure 4—figure supplement 2.

Reviewer 2

*[…] Subsection “CryoEM structure determination” paragraph one. Citing the two papers where amphipol was successfully used would not be enough, especially the journal is targeted toward diverse audience (and also because these two papers do not even cite the original Jean-Luc Popot 1996 PNAS paper on amphipol!).*

We have included a reference to the Popot, 2010 review and to other papers from Popot and colleagues.

*Paragraph two of the same section. I suppose that disulfide-linked AcrAB recruits not only TolC, but also additional AcrA monomers, as seen in Figure 1—figure supplement 2.*

Based on the structure, only three AcrAs form a disulphide bond with the AcrB trimer, and the other 3 AcrAs with single-cysteine mutation in the assembly will be free, and not disulphide linked with AcrB or each other. (The AcrA and AcrB with single-cysteine mutations are overexpressed in an acrAB null strain, so only single-cysteine mutants will be present in the assembly). We have now made it more explicit in the revised text that there are free subunits in the cysteine engineered pump.

*In the last line of paragraph two: I believe that this should be Figure 1—figure supplement 2.*

We have changed this.

*Even with the non-intuitive naming convention of supplementary figures, it does not help that Figure 3 supplement comes before Figure 2 supplement. Please fix this confusing numbering system for the supplementary figures.*

In the revised version, we swapped the order of the Figure 2 and Figure 3, and the numbers for the corresponding supplement figures have also been changed.

*Subsection “The structure of pump in complex with inhibitor MBX3132 reveals the interaction interfaces between pump components at atomic resolution”. Inhibitor MBX3132 gave such wonderful results precisely because it binds very tightly to the binding pocket, as seen in the very large negative values of its calculated binding energy in Sjuts et al., 2016. The high affinity of this compound must be mentioned somewhere.*

We have changed the text to read:

“The inhibitor MBX3132 likely binds AcrB tightly, since it is active in the nanomolar concentration range (Sjuts et al., 2016), and the compound was used to lock the pump in a more homogeneous conformation.”

*In paragraph three of the same section: conserved Val-Gly-Leu/Thr element in AcrA? It may be obvious, but it would be better to specify AcrA because TolC is also discussed in the same paragraph.*

We have changed the text here to read:

“These interactions account for the importance of the conserved Val-Gly-Leu/Thr element of TolC “

*Also in this paragraph, the tip-to-tip interaction of AcrA and TolC HTH motifs is made very clear with Figure 4. However, non-specialists will miss the absence of discussion (and especially figure!) on what happens to these residues in the complex containing a closed TolC conformer. Also, the closed conformer of TolC is supposed to be stabilized by multiple interactions between coiled coils. Are the tip-to-tip interactions enough to overcome these pre-existing multiple interactions?*

We have added a new Figure 6 that compares the tip-to-tip interactions in the apo- and ligand-bound pumps. Based on the observation that the TolC channel is open in the ligand-bound states, we can conclude that the tip-to-tip interactions are sufficient to overcome the stabilising interactions that keep free TolC in a closed state.

*Paragraph four of the same section and Figure 5. This is very nice and informative, but I miss the comparison with the situation in the apo complex. The next paragraph talks about "overlays", but again there is no figure showing such overlays (especially for the AcrA-TolC interface) even in the supplement.*

We have added a new Figure 5—figure supplement 1 and accompanying text that compares the apo- and ligand-bound pumps. These are also shown in the accompanying Video 2.

*Discussion section. If TolC-AcrA interface does not change significantly by the binding of the substrate to AcrB, how could the TolC gate become open in the normal operation of the pump? This reminds us that the "open" complex was seen only when MBX3132 was bound to all three of the trimers, and could be an artifact of the experimental setup.*

It is not true that the open complex was seen only in the presence of the inhibitor; it is also seen in the presence of substrate puromycin. The interfaces are similar in the substrate-bound and inhibitor-bound pumps with an opened TolC channel. In the transport-state, the interface does not change significantly as AcrB cycles through the three conformations, LTO. However, the TolC-AcrA interface does change significantly between the apo and substrate-bound pumps, with a closed and opened TolC channel, respectively.

In the switch from the apo- to the ligand-bound states, the conformational changes in AcrA are effectively quaternary structural adjustments. These entail repacking of the AcrA helical hairpin domains between adjacent protomers in the tube-like portion of the pump, and repacking of the membrane proximal and β-barrel domains with the surface of AcrB. We think that ligand binding by AcrB triggers changes in its periplasmic headpiece, and this in turn changes the interaction surface with the AcrA membrane proximal and β-barrel domains that results in a rotation movement of those domains.

*Reviewer 3*

*[…] AcrZ seems to be present in the structures obtained in the presence of puromycin and in that in the presence of the inhibitor but absent in the apo states. The authors do not comment upon that point. Could it be that indeed AcrZ is recruited differently whether the substrate is present or not? What could be the possible role of AcrZ on the transport and/or activity?*

Yes, it is correct that AcrZ is present in the puromycin and inhibitor bound structures but not in the apo-structure. AcrZ is a modulator of AcrB for the efflux of substrate puromycin. We used the his- tagged AcrZ to pull down the full assembly to prepare the ligand-bound structures. We did not co- express AcrZ with the disulphide-stabilised construct. We have changed the text to make this clearer. AcrZ may not be recruited differently depending on the presence of ligand, based on comparisons of our crystal structure of AcrZ/AcrB complex in the absence of ligand (Du et al., 2014) and the AcrZ/AcrB/puromycin complex reported here. The functional role of AcrZ in the complex is still unclear.

*Note that we advise the authors not to color AcrZ in purple because it is difficult (in particular for a daltonian reviewer) to see it in the very middle of an AcrB molecule colored in blue!*

*Subsection “The transport state of the full pump with opened channel” first paragraph: Could the authors give the reference to the paper mentioning that puromycin is a « validated transport substrate » of AcrB ?*

The reference is Hobbs et al., (2012).

*We believe that here and there, additional papers could be mentioned, see below:*

Subsection “CryoEM Structure determination”: There are more and more papers showing that amphipols are indeed very useful in the EM field. The authors refer to the work by Baker, Fan and Serysheva, 2015 and by Liao, Cao, Julius and Cheng, 2013. We suggest to also cite the following review by J-L Popot (Zoonens and Popot, 2014), so that the lab where this molecule was developed is also acknowledged.

*Subsection “The structure of pump in complex with inhibitor MBX3132 reveals the interaction interfaces between pump components at atomic resolution”, paragraph two: Maintenance of TolC thanks to the interprotomer network involving R367, T152, D153 and Y362 was already shown thanks to electrophysiology (Andersen et al., 2002), microbiology (Augustus et al.,2004) and crystallography (Bavro et al., 2008).*

*Paragraph four of the same section: Non equivalent interactions between each AcrB promoter and each pair of AcrA promoter is reminiscent to the work by the laboratory of Helen Zgurskaya on the related pump TriABC-OpmH (Ntreh, Weeks, Nickels and Zgurskaya, 2016)*

*In paragraph five of the section: Wang et al. comment on the fact that the pump is a highly allosteric system in which conformational changes associated with ligand binding by AcrB are communicated to TolC. Conversely, it was shown that the MFP is mandatory for the activity of the pump (Zgurskaya and Nikaido, 1999), a fact that was nicely discussed by Fischer and Kandt (in section 4.3.3 of Fischer and Kandt, 2013, BBA – Biomembranes 1828, 632-641).*

We have included these references in the modified text.